# Single cell genome sequencing of laboratory mouse microbiota improves taxonomic and functional resolution of this model microbial community

**Svetlana Lyalina**[1], **Ramunas Stepanauskas**[2], **Frank Wu**[1], **Shomyseh Sanjabi**[1,3], **Katherine S. Pollard**[1,4,5]*

**1** Gladstone Institutes, San Francisco, CA, United States of America, **2** Bigelow Laboratory for Ocean Sciences, East Boothbay, ME, United States of America, **3** Department of Microbiology & Immunology, University of California, San Francisco, San Francisco, CA, United States of America, **4** Department of Epidemiology & Biostatistics, Institute for Human Genetics, and Institute for Computational Health Sciences, University of California, San Francisco, San Francisco, CA, United States of America, **5** Chan-Zuckerberg Biohub, San Francisco, CA, United States of America

* katherine.pollard@gladstone.ucsf.edu

**Data Availability Statement:** Genome assemblies and feature annotations are available in a figshare repository (DOI: 10.6084/m9.figshare.c.4454150).

## Abstract

Laboratory mice are widely studied as models of mammalian biology, including the microbiota. However, much of the taxonomic and functional diversity of the mouse gut microbiome is missed in current metagenomic studies, because genome databases have not achieved a balanced representation of the diverse members of this ecosystem. Towards solving this problem, we used flow cytometry and low-coverage sequencing to capture the genomes of 764 single cells from the stool of three laboratory mice. From these, we generated 298 high-coverage microbial genome assemblies, which we annotated for open reading frames and phylogenetic placement. These genomes increase the gene catalog and phylogenetic breadth of the mouse microbiota, adding 135 novel species with the greatest increase in diversity to the *Muribaculaceae* and *Bacteroidaceae* families. This new diversity also improves the read mapping rate, taxonomic classifier performance, and gene detection rate of mouse stool metagenomes. The novel microbial functions revealed through our single-cell genomes highlight previously invisible pathways that may be important for life in the murine gastrointestinal tract.

## Introduction

The number of microbial species with at least one genome sequence has grown rapidly in recent years. The human gut has been a major focus of these efforts [1–5], with metagenome assembled genomes (MAGs) and innovations in culturing [6–8] capturing genomes for many species previously absent from databases built primarily through isolate sequencing.

Mice are a model system for host-associated microbiota. They are heavily utilized in biomedical research as well as basic science investigations of community assembly and resilience.

**Funding:** This work was funded by the Chan Zuckerberg Biohub Investigatorship (https://www.czbiohub.org/) in the form of funds to KSP. This work was also funded by Gladstone Institutes in the form of grants to RS [NSF grant 1826734, NSF grant OCE-1335810, NIH grant R21AI134037], and to SS and KSP [NIH grant R21AI108953].

**Competing interests:** The authors have declared that no competing interests exist.

However, the species present in wild and laboratory mouse stool are heavily under-represented in genome databases in comparison to human-associated microbiota [9]. This gap can create a biased picture of the functional and taxonomic landscape of shotgun metagenomic studies carried out in mice, since most bioinformatics methods rely on available reference data. Several research groups have actively sought to address this problem, both by focusing on mouse-specific bacterial strains that were previously unculturable [10] and by performing co-assembly of large-scale metagenomic datasets from a broad variety of mouse facilities [11].

This study aims to increase the number of mouse gut species with a sequenced genome using microbial single-cell genomics (SCG). Our workflow leverages fluorescence-activated cell sorting (FACS), whole genome amplification with WGA-X, shotgun sequencing and *de novo* assembly of genomes from individual microbial cells from two laboratory mouse strains [12]. By annotating the taxonomy and encoded functions of 298 quality-controlled, single-cell genomes, we revealed previously invisible pathways and phylogenetic breadth, increasing the power of metagenomic analysis tools. These results demonstrate the utility of SCG for characterizing host-associated microbiomes and provide a resource towards a better understanding of the mouse gut as a model system.

## Results

The biological material used for this study came from fecal pellets of three mice of two different strains—two wild-type C57BL/6N mice and a transgenic CD4-dnTβRII (DNR) mouse prone to developing intestinal inflammation [13]. These two strains' intestinal microbiota have been previously studied within the lab [14], which allowed us to evaluate how the single-cell genomes we produced change previous interpretations of shotgun metagenomic data.

Using stool from these mice, we performed FACS followed by whole genome amplification with WGA-X. Cell sorting was based on the fluorescence of nucleic acids stain SYTO-9 (Thermo Fisher Scientific) and light scatter signals using a previously established gate for individual prokaryotic cells [12]. To assess the general structure of the microbiomes, we first performed low-coverage sequencing and assembly of 738 cells (median 765,918 reads/sample [342,424 – 2,670,861], median coverage 21.7 [18.5–26.9]) (Methods). We filtered the resulting single-cell amplified genomes (SAGs) to exclude assemblies with total length below 20,000 basepairs (bp) or suspected to be contaminated (determined by nucleotide tetramer principal components analysis [15]), producing 697 SAGs that vary in quality and completeness (Fig 1). Compared to the earlier, multiple displacement amplification (MDA) technique [16], the WGA-X approach has been shown to improve the amplification of single-cell DNA, especially for microorganisms with high GC-content genomes [12], and we indeed observed a wide range of GC% across the assemblies (Fig 1E). This advantageous property of WGA-X has been demonstrated in prior work on environmental microbiomes, and we hypothesize that this study also benefits from the improved performance (based on the demonstrated GC range). However, it is important to note that the genomes of host-associated microbes generally have less extreme GC% values [17] and therefore a replicate experiment with the standard MDA technique would be required to conclusively show this.

We next selected two samples, one of each strain, for further sequencing towards obtaining high-coverage SAGs. To prioritize cells that would produce high-quality data and increase the taxonomic diversity of mouse gut genomes, we performed phylogenetic placement of the low-coverage SAGs with GTDB-Tk [18], successfully placing 448 SAGs within the GTDB genome tree of life [19] (release 86). We then selected the 150 SAGs from each sample that maximize phylogenetic diversity and excluded SAGs with low probability of high genome recovery (Methods). Further sequencing and assembly of DNA from the corresponding cells produced

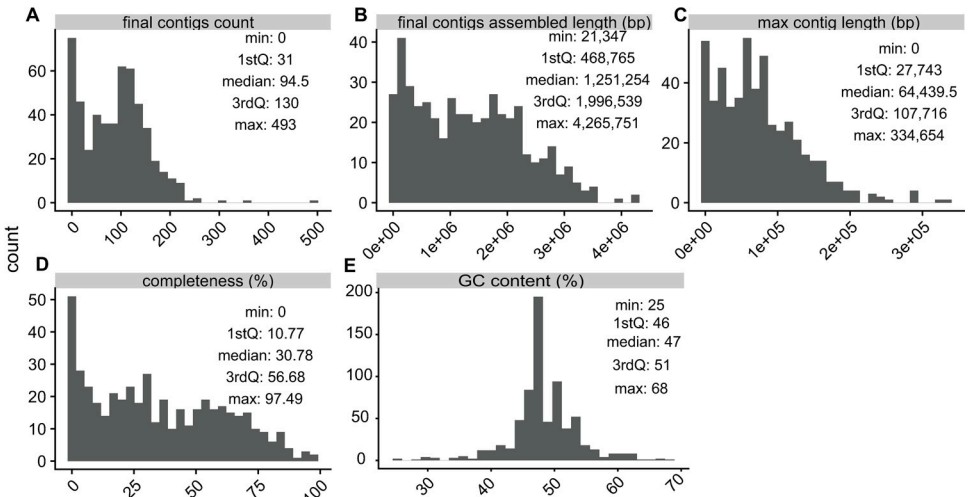

**Fig 1. Quality metrics of low-coverage SAG assemblies.** A faceted plot containing histograms of quality metrics used to describe the assembled SAGs. The facets display the following metrics: A) total number of contigs, B) their total assembled lengths (in number of nucleotide basepairs), C) the length of the longest contig in each assembly (in number of nucleotide basepairs), D) CheckM estimated completeness (as percentage), and E) GC content. Tukey five-number summaries (minimum, 25% quantile, median, 75% quantile, maximum) are overlaid on each metric's panel.

298 high-coverage SAGs after quality control (median 2,918,599 reads/cell [2,361,607 – 3,637,757]; median coverage 30.6 [27.9–32.8]). Two SAGs were discarded due to the high likelihood of containing DNA from multiple cells, which was determined via a combination of tetramer PCA outlier identification and blastn [20, 21] against the NCBI nt database [22]. As expected, these final 298 SAGs show significant improvements in relevant quality metrics when compared to corresponding low-coverage assemblies (S1 Fig). All subsequent analyses use the high-coverage SAGs.

To evaluate whether the SAGs increased the diversity of sequenced mouse microbiota, we placed them on the GTDB tree and quantified the additional branch length added by SAGs compared to the total branch length from previously sequenced microbial samples. This metric is further referred to in the text as phylogenetic gain. Evaluating this metric across clades, we observed that our SAGs primarily increase the phylogenetic diversity of the *Muribaculaceae* and *Bacteroidaceae* families (Fig 2). Despite the fact that GTDB includes MAGs from uncultured microbes, this study adds substantial new diversity to the tree, with 135 out of 298 SAGs having no hit in the GTDB release 86 with FastANI [23]-computed average nucleotide identity above 97%.

Next, we investigated the gene content of the SAGs. We annotated open reading frames in all SAGs, dereplicated these, and analyzed their functional potential using annotations from clusters of orthologous groups (COGs) [24]. Gene sequences were evaluated for percent nucleotide identity to all sequences in a previously published mouse stool metagenome-derived gene catalog (4) and labeled as novel if they have no matches above 95% nucleotide identity. Overall, 53.7% of SAG genes were novel and 46.3% overlapped with the mouse catalog, which compares to 10% overlap with a human gene catalog and <0.1% for a marine catalog (Fig 3), highlighting the functional differences of microbes across these environments. Novel SAG genes were enriched for COG categories M (Cell wall/membrane/envelope biogenesis), L (Replication, recombination and repair), C (Energy production and conversion) and R (General function prediction only). This enrichment was determined by Annotation Enrichment Analysis [25], a method that aims to reduce the bias towards highly annotated functional

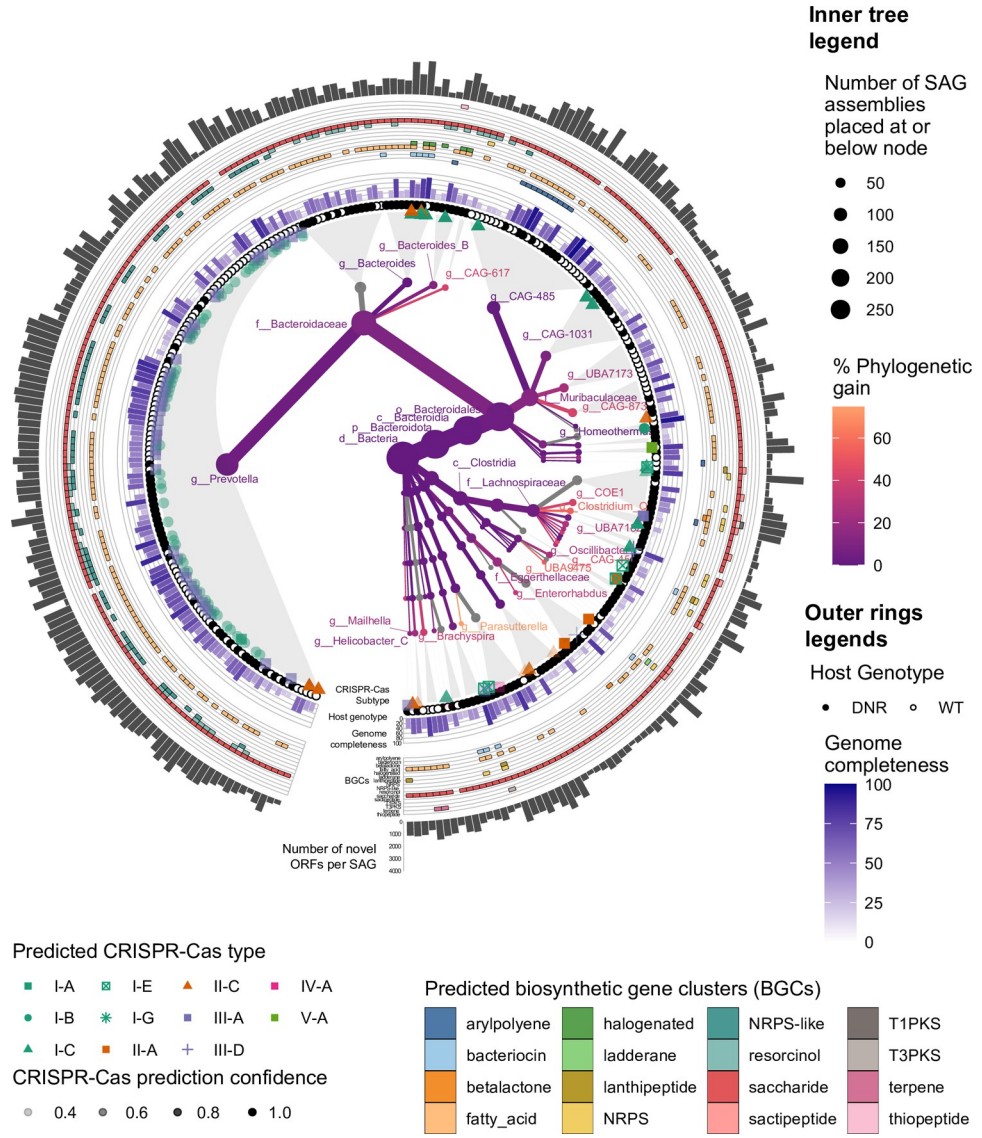

**Fig 2. SAGs increase phylogenetic diversity and contain distinct genomic features.** The central part of this circular figure contains a heat tree reflecting the number of SAG assemblies placed at different sub-branches of the GTDB v86 bacterial genome tree (represented by node size), and percentage phylogenetic gain achieved by the insertion of the new genome assemblies (represented by color scale). The outer rings of the figure contain additional genomic feature information inferred about the successfully placed SAG assemblies. Proceeding radially outwards, the additional markings denote predicted CRISPR-Cas system type and the confidence of this prediction (ring of single point symbols of variable opacity), estimates of genome completeness (ring of barplots), the number of genes contributing to predicted biosynthetic gene clusters (outermost ring of colored polygons), and number of open reading frames contributing novel entries to the gene catalog when compared against a previously published resource (outer ring of barplots).

categories and utilize the hierarchical structure in a given functional ontology. While these annotation categories provide a rather broad summary of the functions distinct to this gene set, they generally suggest that sequencing more members of the microbiota would expand our understanding of both internal housekeeping functions (categories L and R), but also func- tions more pertinent for translational applications within category M, which contains potential candidates for studying interactions with the host immune system. Thus, our SAG gene

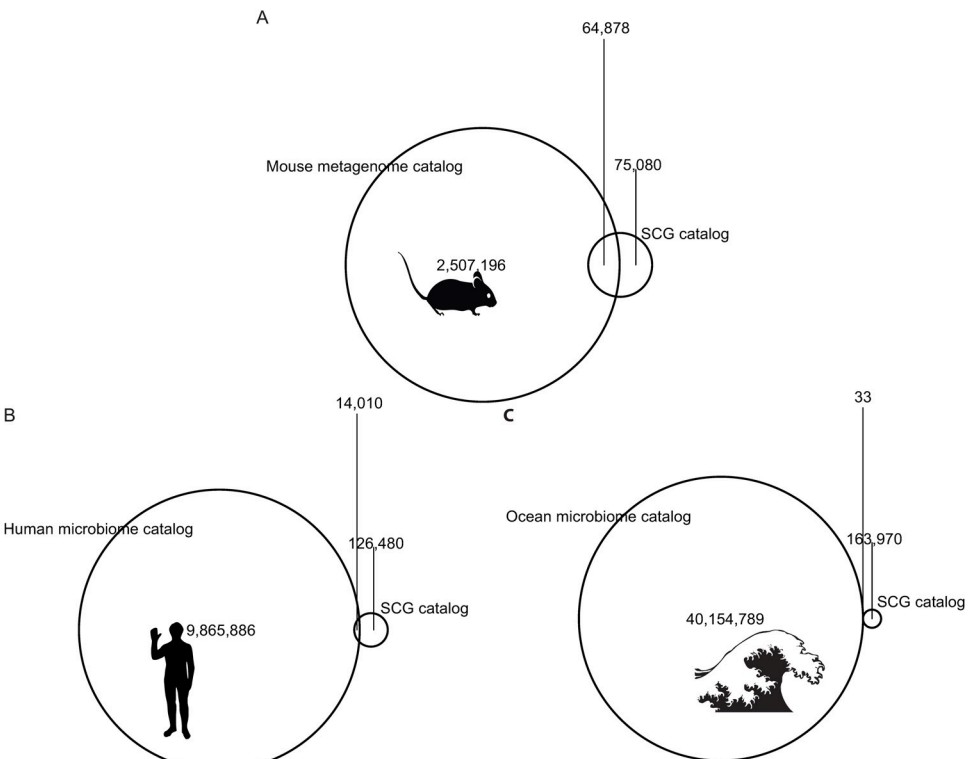

**Fig 3. A gene catalog derived from SAGs shows substantial novelty when compared against other microbiome gene catalogs.** Venn diagrams reflect the shared and unique counts of genes when comparing the set of non-redundant genes from this study's data against previously published gene catalogs derived from metagenomic sequencing efforts in A) mice, B) humans, and C) marine samples.

catalog expands the representation of putative functions present in mouse gut microbes, with surprisingly large gains given the number of genomes sequenced for this study.

To expand beyond COG annotations for two important groups of genes, we performed additional annotation of enzymes involved in secondary metabolism and CRISPR-associated (Cas) proteins along with their CRISPR arrays. Overall, 3,257 putative secondary metabolism gene clusters were found across the 298 SAGs sequenced at high coverage. The most prevalent predicted cluster types were the broad categories of saccharide, fatty acid, and NRPS-like, whereas the more nuanced product types were detected much more rarely. CRISPR-Cas types were determined in 89 genomes, of which 22 genomes had 2 CRISPR complexes on different contigs. Amongst these genomes, we observed some heterogeneity in CRISPR-Cas types across phylogenetic groups (Fig 2), which is expected based on isolate genomes. An additional 31 genomes had Cas operons, but no proximal CRISPR array. To test the hypothesis that failure to find CRISPR arrays in many genomes stems from lower completeness we performed a Wilcoxon rank sum test between the CheckM estimated completeness measures of these two sets of genomes (median of "Cas operon + CRISPR array" group was 60.42% vs that of the "Cas operon + no CRISPR array" group was 40.29%). The test was significant with a p-value of 0.001425, suggesting the "no array" group does in fact contain lower quality assemblies.

The distributions of biosynthetic gene clusters (BGCs) and CRISPR-Cas systems in our SAGs support the phylogenetic novelty of several clades characterized in this study. We quantified the presence of BGCs and CRISPR-cas types in relation to the phylogenetic placement of the contributing genome (outer ring of Fig 2). In the context of this trimmed genome subtree,

the newly sequenced *Prevotella* SAGs form a qualitatively distinct, relatively flat phylogenetic subcluster, distinguished by unique CRISPR-Cas subtype patterns and presence of NRPS-like predicted BGCs. A closely related subset of SAGs assigned to the genus CAG-486 within the *Muribaculaceae* family accounts for a high proportion of identified aryl polyene BGCs, suggesting similar adaptations to oxidative stress [26]. Thus, the new taxonomic diversity we captured is mirrored by gene functional profiles that differ from related genomes.

Finally, we investigated to what degree our SAGs improve the sensitivity and resolution of metagenomic analysis using 236 shotgun metagenome samples from laboratory mouse stool, as well as metagenomes from wild mouse stool (N = 10), human stool (N = 274), and marine environments (N = 20, subset of full data) (accessions listed in S1 Table). Focusing on taxonomic classifiers, we created custom mapping references for sourmash [27] and MIDAS [9], which represent two common approaches: kmer-based versus marker gene-based. We compared taxonomic coverage and prevalence estimates with each tool using the database distributed with the software, a database composed only of SAGs, and the two combined. For both tools, the combined database generally improved the taxonomic classification of mouse microbiome samples, with the exception of the wild mouse microbiome testing scenario, which only showed improvement with FDR < 0.1 when using sourmash and not MIDAS (Fig 4). Interestingly the addition of SAGs also improved classification rates to a limited degree with human microbiome samples (paired test on sourmash results only), but not marine samples. The sourmash results should not be directly compared with those of MIDAS as the input data and match stringency differ between them in two important ways: (1) sourmash (like similar Min-Hash-based tools) operates on representative compressed subsets of the original sequence's full set of kmers; (2) sourmash is able to assign kmer hashes to higher levels of the taxonomy (up to and including the domain level), while MIDAS is constrained to species clusters. The full results of non-parametric testing of the performance of pairs of databases for each dataset and tool type can be found in S2 Table, with highlighted rows showing cases of significant performance improvement in a number of host-associated shotgun microbiome datasets. Ridgeline plots graphically portray these performance differences in greater detail (S2 and S3 Figs), expanding upon the information shown in Fig 4. These results show that the novel phylogenetic diversity we captured with SAGs has a positive effect on our ability to taxonomically profile shotgun metagenomes from the mammalian gut.

## Discussion

To our knowledge, this study is the first to generate single-cell genome assemblies from mammal-associated microbiota with the WGA-X approach. The draft genomes that we assembled increase the phylogenetic diversity of mouse gut microbiota in public databases. Our SAGs add a particularly large number of genomes (58 assemblies) to the recently proposed candidate family Muribaculaceae within the Bacteroidales, previously referred to in the literature as S24-7 and Ca. Homeothermaceae [28, 29]. This family has been reported as a taxon of interest in multiple studies [30–32] but has so far only been characterized via 16S markers and MAGs. Only one recent paper has successfully isolated members of this family in culture [29]. Another taxon with large numbers of newly placed SAGs (120 assemblies), though small phylogenetic gain (4.27%; defined as the percentage increase in total branch length arising from the addition of new nodes), is the genus *Prevotella*, which contains Gram-negative obligate anaerobes with potential links to mucosal inflammation susceptibility [33]. *Prevotella* were also associated with mice being fed a high fat diet in the study of Xiao et al. [11], another experimental condition that can lead to increased inflammation. The Lachnospiraceae family had a comparable percentage of phylogenetic gain (4.57%) to the *Prevotella* genus, although this was achieved

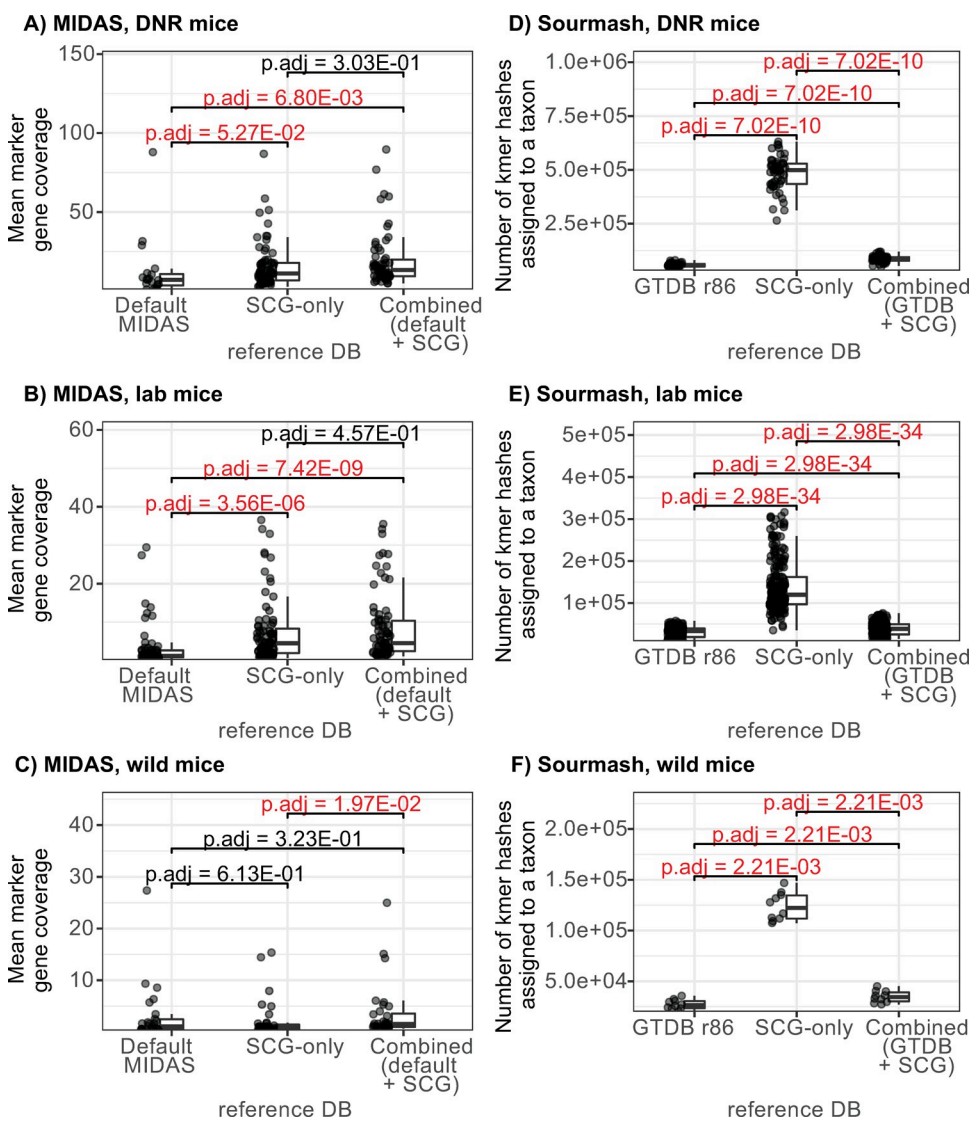

**Fig 4. Taxonomic classifiers perform better on murine metagenomic samples when their reference databases are augmented with SCG data.** Boxplots and swarmplots show the performance metrics of two classifiers (left column—mean marker gene coverage by MIDAS, right column—total number of kmer hashes assigned by sourmash) on three test metagenomic inputs (1st row—samples from DNR mice, 2nd row—samples from other lab mice, 3rd row—samples from wild mice). Brackets over the boxplots display FDR-adjusted (Benjamini-Hochberg procedure) p-values from pairwise comparisons with the unpaired (MIDAS) and paired (sourmash) Wilcoxon rank-sum test. Text over significance brackets is colored red in cases where the adjusted p-value is less than 0.1.

with a smaller number of new SAGs (25), highlighting the difference between raw number of taxonomic additions and their contribution to phylogenetic diversity. Overall this family was 3rd in number of new SAGs placed within it, following Bacteroidaceae (154) and the aforementioned Muribaculaceae. Only 13 of the putative Lachnospiraceae SAGs received a taxonomy prediction down to the genus level, hindering more detailed hypotheses about the influence and origins of these community members. The literature suggests that Lachnospiraceae members of the mammalian gut microbiome aid in the production of anti-inflammatory short chain fatty acids [34], although not all isolates were shown to have this ability. This is also suggested by our data, as our biosynthetic gene cluster predictions for this clade do not show a

universal ability to produce fatty acids. Similar to the dialkylersorcinol biosynthesis result discussed in a later section, we also see that the Lachnospiraceae SAGs are primarily found in the sample taken from the DNR mouse (21 out of 25 Lachnospiraceae SAGs), which suggests an intriguing relationship between the mouse genetic background (prone to inflammatory bowel disease) and observed lack of inflammation. In summary, our SAGs add genomes for important taxonomic groups in the mouse microbiota, although this phylogenetic novelty is necessarily constrained by the limitations of a host-associated, laboratory environment.

SAGs also increase our knowledge of the functional potential of microbes in the mouse gut. Gain in functional novelty includes a large number of COGs that were enriched and depleted compared to open reading frames previously observed in mouse stool samples. When summarizing these differentially detected functional categories, four are particularly enriched: energy production and conversion (C), replication and repair (L), cell wall/membrane/envelope biogenesis (M), and the unspecific category (R)—general function prediction only. Previously unobserved sequences classified under the M category could be of interest when mining for new antigenic proteins, whereas genes placed in the unspecific R category could be further experimentally probed to shed light on microbial "dark matter".

Our annotations of SAGs for secondary metabolism genes and CRISPR systems aim to highlight the capacity of this sequencing approach to more faithfully reflect intra-genome structure. When analyzed in the context of phylogenetic relationships between SAGs, the results of CRISPR-Cas type identification show SAGs placed in the *Prevotella* genus have both Type I and Type III systems, whereas this is relatively uncommon in our data outside this clade. This suggests that these microbes have a more sophisticated defense repertoire that allows for targeting of both DNA and RNA [35]. These findings should be viewed with a critical eye, however, as predictions for this group of genomes had lower confidence scores reported by the CRISPR-Cas prediction algorithm, possibly due to overall dissimilarity between these new sequences and the subtyping tool's training data. Variable genome completeness could be another confounding factor limiting both BGC reconstruction and CRISPR-Cas identification. Despite these limitations, systematic cataloging efforts expanding upon our current study could in the future be used to investigate histories of bacteria-phage interactions or spearhead bio-prospecting efforts to uncover novel genome editing tools.

Looking at secondary metabolism, we see that the most widely represented gene clusters are for saccharide and fatty acid biosynthesis. The remaining categories are sparsely observed. An interesting clustering occurs for the resorcinol group which appears primarily to be present in genomes from the Bacteroidaceae family. This cluster type originates mainly from genomes found in the DNR mouse microbiome (34 resorcinol clusters predicted, vs only 6 from WT). The particular gene that is considered by the predictive tool AntiSMASH as a signature gene for the resorcinol annotation is DarB (KEGG orthology ID of K00648), which falls under the fatty acid biosynthesis KEGG pathway. The literature provides limited insight into what microbiome activities resorcinol biosynthesis could be relevant to, however, some reported associations of the more specific chemical family of dialkylresorcinols include anti-inflammatory, anti-proliferative, and antibiotic activities [36]. Interestingly, a dialkylresorcinol compound has been used to attenuate the effects of experimentally induced intestinal inflammation [37], which has potential implications for the observed higher prevalence of dialkylresorcinol-producing genomes in the inflammation-prone DNR mouse strain.

Considering the relatively modest costs of this sequencing experiment, we were surprised to find that the new sequences significantly helped with metagenomic read recruitment even in unrelated mouse lines and wild mouse samples, which have been shown to have more diverse microbiomes than their laboratory counterparts [38]. This corroborates prior reports demonstrating the value of SAG genomes as reference material for the interpretation of marine

[39, 40] and soil [12, 41] microbiome omics data. The lack of improvement of the taxonomic classifiers on marine metagenomic data with mouse microbiome SAGs agree with our findings of novel genes, confirming the lack of highly similar genomes between these two environments.

Despite single-cell sequencing being a promising approach for increasing the representation of unculturable mouse symbionts in the tree of life, certain caveats still exist. For example, although the individual SAG assemblies have acceptable quality metrics, there is a limit to the completeness that can be achieved when operating with short read sequencing data. Long repetitive segments continue to pose an obstacle to assemblers that attempt to span ambiguous regions of the genome. Whole genome amplification, while drastically improved by the WGA-X process, is still not uniform across the genome, thus requiring a relatively deep sequencing of SAGs in order to access under-amplified regions. Finally, the inherent upper bound on observable diversity in a small number of ecological sites (just two individual mice in the present study) means that a much broader sample acquisition effort needs to be undertaken, spanning more mouse strains, geographic locations, diets, and other experimental perturbations. Despite these limitations, we expect that the taxonomic and functional novelty revealed in this study will encourage others to leverage single-cell genomics technologies, as the informational gains observed are only the beginning of what can be uncovered.

## Materials and methods

### Sample acquisition and sequencing

Cells were sequenced from three murine fecal pellets, two from wild-type C57BL/6N mice and one from an inflammatory bowel disease model CD4-dnTGFBRII (DNR) [13, 42] mouse not exhibiting intestinal pathology at the time of sampling. To preserve the mouse feces, a cryo-preservation "glyTE" stock (11.11x) was made by mixing 20 mL of 100x Tris-EDTA pH 8.0 (Sigma) with 60 mL deionized water and 100 mL molecular-grade glycerol (Acros Organics). This mixture was filter-sterilized using a 0.2 micrometer filter. Prior to use, 1x glyTE was made by diluting with phosphate buffered saline (PBS) at a 10:1 ratio. 1 mL of the 1x glyTE was then aliquoted into cryotubes. Each fecal pellet was distributed into 3 separate cryotubes to create 3 replicates for each sample. Each sample was dispersed into the solution by gentle pipetting and allowed to incubate at room temperature for 1 minute before being placed on dry ice. Samples were stored at -80 C and shipped on dry ice to the Bigelow Laboratory's Single Cell Genomics Center for further processing using a previously described protocol [12], the details of which are summarized in the following paragraphs.

After thawing, field samples were incubated with the SYTO-9 DNA stain (5 μM; Thermo Fisher Scientific) for 10–60 min. Fluorescence-activated cell sorting (FACS) was performed using a BD InFlux Mariner flow cytometer equipped with a 488 nm laser for excitation and a 70 μm nozzle orifice (Becton Dickinson, formerly Cytopeia). The cytometer was triggered on side scatter, and the "single-1 drop" mode was used for maximal sort purity. Sort gate was defined based on particle green fluorescence (proxy to nucleic acid content), light side scatter (proxy to size), and the ratio of green versus red fluorescence (for improved discrimination of cells from detrital particles). Cells were deposited into 384-well microplates containing 600 nL per well of 1x TE buffer and stored at −80˚C until further processing. Of the 384 wells, 317 wells were dedicated for single cells, 64 wells were used as negative controls (no droplet deposition), and 3 wells received 10 cells each to serve as positive controls. The accuracy of droplet deposition into microplate wells was confirmed several times during each sort day, by sorting 3.46 μm diameter SPHERO Rainbow Fluorescent Particles (Sperotech Inc.) and microscopically examining their presence at the bottom of each well. In these examinations, <2% wells

did not contain beads and <0.4% wells contained more than one bead. Index sort data were collected using the BD FACS Sortware software. The following laboratory cultures were used in the development of a cell diameter equivalent calibration curve: *Prochlorococcus marinus* CCMP 2389, *Microbacterium* sp., *Pelagibacter ubique* HTCC1062, and *Synechococcus* CCMP 2515. Average cell diameters of these cultures were determined using Mulitisizer 4e (Beckman Coulter). Average light forward scatter of each of the four cultures was determined using the same BD InFlux Mariner settings as in environmental sample sorting and was repeated each day of single cell sorting. We observed a strong correlation between cell diameters and light forward scatter (FSC) among these cultures [12]. Taking advantage of this correlation, the diameter equivalent of the sorted environmental cells (D) was estimated from a log-linear regression model: D = 10^(a * log10(FSC)—b), where a and b are empirically derived regression coefficients [12].

Prior to genomic DNA amplification, cells were lysed and their DNA was denatured by two freeze-thaw cycles, the addition of 700 nL of a lysis buffer consisting of 0.4 M KOH, 10 mM EDTA and 100 mM dithiothreitol, and a subsequent 10-minute incubation at 20˚C. The lysis was terminated by the addition of 700 nL of 1 M Tris-HCl, pH 4. Single cell whole genome amplification was performed using WGA-X. Briefly, the 10 μL WGA-X reactions contained 0.2 U μL-1 Equiphi29 polymerase (Thermo Fisher Scientific), 1x Equiphi29 reaction buffer (Thermo Fisher Scientific), 0.4 μM each dNTP (New England BioLabs), 10 μM dithiothreitol (Thermo Fisher Scientific), 40 μM random heptamers with two 3'-terminal phosphorothioated nucleotide bonds (Integrated DNA Technologies), and 1 μM SYTO-9 (Thermo Fisher Scientific) (all final concentrations). These reactions were performed at 45˚C for 12–16 h, then inactivated by a 15 min incubation at 75˚C. In order to prevent WGA-X reactions from contamination with non-target DNA, all cell lysis and DNA amplification reagents were treated with UV in a Stratalinker (Stratagene) [43]. An empirical optimization of the UV exposure was performed in order to determine the length of UV exposure that is necessary to cross-link all detectable contaminants without inactivating the reaction. Cell sorting, lysis and WGA-X setup were performed in a HEPA-filtered environment conforming to Class 1000 cleanroom specifications. Prior to cell sorting, the instrument, the reagents and the workspace were decontaminated for DNA using UV irradiation and sodium hypochlorite solution [44]. To further reduce the risk of DNA contamination, and to improve accuracy and throughput, Bravo (Agilent Technologies) and Freedom Evo (Tecan) robotic liquid handlers were used for all liquid handling in 384-well plates.

Libraries for SAG genomic sequencing were created with Nextera XT (Illumina) reagents following manufacturer's instructions, except for purification steps, which were done with column cleanup kits (QIAGEN), and library size selection, which was done with BluePippin (Sage Science, Beverly, MA), with a target size of 500±50 bp. DNA concentration measurements were performed with Quant-iT™ dsDNA Assay Kits (Thermo Fisher Scientific), following manufacturer's instructions. Libraries were sequenced with NextSeq 500 (Illumina) in 2x150 bp mode using v.2.5 reagents. The obtained sequence reads were quality-trimmed with Trimmomatic v0.32 [45] using the following settings: -phred33 LEADING:0 TRAILING:5 SLIDINGWINDOW:4:15 MINLEN:36. Reads matching the *Homo sapiens* reference assembly GRCh38, *Mus musculus* reference assembly mm10, and a local database of WGA-X reagent contaminants (≥95% identity of ≥100 bp alignments), as well as low complexity reads (containing <5% of any nucleotide) were removed. The remaining reads were digitally normalized with kmernorm 1.05 (http://sourceforge.net/projects/kmernorm) using settings -k 21 -t 30 -c 3 and then assembled with SPAdes v.3.9.0 [46] using the following settings:—careful—sc—phred-offset 33. Each end of the obtained contigs was trimmed by 100 bp, and then only contigs longer than 2,000 bp were retained. Contigs matching the *H. sapiens* reference assembly

GRCh38 and a local database of WGA-X reagent contaminants (≥95% identity of ≥100 bp alignments) were removed. The quality of the resulting genome assemblies was determined using CheckM v.1.0.7 [47] and tetramer frequency analysis [15]. This workflow was evaluated for assembly errors using three bacterial benchmark cultures with diverse genome complexity and %GC, indicating no non-target and undefined bases in the assemblies and average frequencies of mis-assemblies, indels and mismatches per 100 kbp: 1.5, 3.0 and 5.0 [12].

Functional annotation was first performed using Prokka v1.12 [48] with default Swiss-Prot databases supplied by the software. Prokka was run a second time with a custom protein annotation database built from compiling Swiss-Prot entries for Archaea and Bacteria.

Low-coverage SAG assemblies were initially generated to evaluate microbiome composition. Two samples, one of each murine host genotype, were selected for follow-up high-coverage sequencing—no discernable differences were observed between the two WT mice, hence one was chosen at random. In each sample, cells were prioritized for high coverage sequencing by optimizing for robust amplification profiles and maximizing the phylogenetic diversity (python code DOI: 10.5281/zenodo.2749707). This approach aims to avoid choosing cells that are only distinguished by their relative ease of amplification, but may otherwise not be particularly rich in novel information. When originally searching for a solution to this problem we were inspired by the proposed solutions to the species prioritization problem in conservation biology [49, 50], specifically choosing a mixed integer programming approach. The criterion used to assess amplification dynamics was computed as the time needed to reach the inflection point in the amplification curve, which was supplied as the per-cell "cost" to the optimization procedure. These costs were provided to the algorithm along with the approximate phylogenetic tree obtained by placing the low-coverage assemblies in a reference genome tree (see next section for phylogenetic placement tools and data used). The mixed integer programming approach finds the optimal set of nodes in a bipartite graph that maximizes the sum of active branch lengths (phylogenetic diversity [51]) while constraining the sum of costs (times to inflection in each amplification reaction). Using the high coverage sequencing data, raw reads were processed into assembled contigs, which were further filtered to yield sufficient quality SAGs, which were assessed by checkM [47] for contamination and assigned a putative taxonomic lineage. All steps performed on the high coverage data (filtering, assembly, initial functional annotation) were the same as described above for low coverage sequences.

## Computational analyses of phylogenetic placement and predicted gene function

We used pplacer v1.1.alpha19 [52] within GTDB-Tk v0.1.3 [18] to phylogenetically place the SAGs in the genome tree that is part of GTDB release 86. The resulting placements of high-coverage SAGs were used to calculate phylogenetic diversity [51] and phylogenetic gain (percent increase in phylogenetic diversity at each tree node) using GenomeTreeTk v0.00.37 [53]. The SAGs were separately assessed for novelty in relation to the genomes in GTDB r86 by computing approximate pairwise average nucleotide identities between them with FastANI v1.0 [23]. The heat tree visualization at the center of Fig 2 was inspired by the approach illustrated in the metacoder [54] R package and was ultimately generated alongside additional genomic feature annotation via the ggtree v3.3.0 [55] and ggtreeExtra v1.5.0 [56] packages, with the overall plot construction and arrangement performed by ggplot2 v3.3.5 [57] and cowplot v1.1.1 [58] packages, and requisite data manipulation facilitated by dplyr v1.0.0 [59] and purrr v0.3.4 [60].

Classification of the CRISPR-Cas system types and subtypes was done by CRISPRCasTyper v1.6.1 [61], which uses HMMER3 v3.3.2 [62], MinCED v0.4.2 [63], Prodigal v2.6.3 [64], and

xgboost v1.5.0 [65] in its predictions. A non-default classification threshold of 0.5 was supplied to generate the broadest set of putative CRISPR-Cas predictions. Identification of secondary metabolism gene clusters was performed with AntiSMASH v5.2 [66]. AntiSMASH uses Prodigal v2.6.3 for gene calling, and HMMER3 v3.2 and BLAST v2.8 for similarity search amongst its curated databases. Default settings were used with AntiSMASH, aside from the addition of '—hmmdetection-strictness loose—allow-long-headers'.

Clustering of Prokka-predicted genes was performed by CD-HIT-EST v4.6.8 [67] (settings:–r 1 –c 0.95 –n 8), and the resulting gene catalog was compared by CD-HIT-EST-2D to previously published gene catalogs derived from mouse [11], human [68], and marine [69] microbiomes. Catalog overlap was visualized using the VennDiagram R package (v1.6.20) [70]. To gauge enrichment of functional categories for novel sequences in our catalog, we annotated the sequences with EggNOG-mapper v1.0.3 [71] using DIAMOND v0.8.36 [72] as the homology search method and then applied Annotation Enrichment Analysis methodology [25] to assess the relationship between the number of genes assigned to a COG [24] category and their novelty in relation to the previously published mouse metagenome catalog [10]. We corrected for multiple testing using the p.adjust function in base R [73] (v3.6.0), using the Benjamini-Hochberg [74] method.

## Comparative analyses of metagenomic read recruitment

Custom sourmash v2.0.0a10 [27] lowest common ancestor (LCA) databases for the set of GTDB genomes and SAG assemblies were created using the "sourmash lca index" function, and metagenomic datasets were then classified with "sourmash lca summarize" using the two databases separately as well as together to evaluate the effect of combining the data. To create the relevant databases for MIDAS v1.3.2 [9], we used the built-in database creation script within the package, as well as an auxiliary step of assigning certain SAG assemblies to pre-existing genome clusters by computing their Mash v2.0 [75] distance to extant cluster representatives. Comparative metagenomic datasets for wild mouse [38], lab mouse [11], human type I diabetes [76], healthy humans [77], and ocean samples [69] were retrieved from the SRA (accession IDs in S1 Table) and converted to fastq with NCBI's fastq-dump utility. Metagenomic datasets from wild-type and DNR mice previously studied at the Gladstone Institutes [14] can be found under BioProject PRJNA397886. We used a paired Wilcoxon rank sum test to evaluate the change in total kmer hash recruitment by sourmash for the three pairs of reference database settings (default vs SAG-only, default vs combination, combination vs SAG-only). We also tested the difference in the number of species that were assigned more than 5 kmer hashes, as an approximation for species prevalence. For MIDAS, we evaluated differences in median and mean coverage of marker genes, as well as the species prevalence, using the unpaired Wilcoxon rank sum test.

## Supporting information

**S1 Table. Accessions used for taxonomic classifier performance evaluation.** Public data retrieved from SRA and ENA to test the performance of metagenomic classifiers with custom reference databases.
(XLS)

**S2 Table. Results of nonparametric comparisons of taxonomic classifier performance with varying reference databases.** Results of Wilcoxon rank sum tests comparing metagenomic read recruitment metrics for every pairwise combination of reference type (default, single-cell genomes only, combined) and test dataset. Two sheets are present in the file, reflecting the

results from two different taxonomic classifiers (sourmash and MIDAS). Tests on sourmash results were paired, while those on MIDAS results were unpaired.
(XLSX)

**S1 Fig. Assembly quality improvement with high coverage sequencing.** Multiple metrics are improved when comparing high coverage versus low coverage single cell sequencing data. Facets show the individual metrics assessed: assembly completeness as determined by CheckM, percentage of reads filtered out as contaminants, total length of the genome assembly in base pairs (bp), maximum contig length (in bp), total number of reads generated. Numbers over each boxplot represent p-values of paired Wilcoxon rank sum tests.
(PDF)

**S2 Fig. Distributions of taxonomic classifier performance metrics when using the taxonomic classifier sourmash and varying reference databases.** Ridgeline plots representing distributions of 2 metagenomic classifier performance metrics when using sourmash—total number of kmer hashes assigned and number of species with more than 5 kmer hashes (an approximation for prevalence). Ridgeline plots are a form of multi-distribution density plot that vertically separate the individual distributions to improve clarity in cases of overlap, necessitating the removal of the traditional y axis label ("density"). This renders the densities not directly comparable between individual ridge lines, but aids in assessment of differences in skew, bimodality, and potential pronounced shift of distribution peaks.The plots are faceted by test metagenomic dataset, and each line within the facet reflects one of the three reference database options—default set of genomes available in GTDB release 86 (labeled "gtdb_ref"), a custom database with single-cell genomes only (labeled "scg_ref"), and a combined database with the GTDB r86 and single-cell genomes (labeled "combined_ref").
(PDF)

**S3 Fig. Distributions of taxonomic classifier performance metrics when using the taxonomic classifier MIDAS and varying reference databases.** Ridgeline plots representing distributions of 3 metagenomic classifier performance metrics when using MIDAS—mean coverage of 15 phylogenetically informative marker genes, median coverage of the same genes, and prevalence (number of samples a species is present in). Ridgeline plots are a form of multi-distribution density plot that vertically separate the individual distributions to improve clarity in cases of overlap, necessitating the removal of the traditional y axis label ("density"). This renders the densities not directly comparable between individual ridge lines, but aids in assessment of differences in skew, bimodality, and potential pronounced shift of distribution peaks. The plots are faceted by test metagenomic dataset, and each line within the facet reflects one of the three reference database options—default MIDAS v1.2 database (labeled "midas_db_v1.2"), a custom database with single-cell genomes only (labeled "midas_db_scg_only"), and a combined database with the MIDAS v1.2 and single cell genomes (labeled "midas_db_combined").
(PDF)

## Acknowledgments

We thank the staff of the Bigelow Laboratory Single Cell Genomics Center for the generation of single cell genomics data.

## Author Contributions

**Conceptualization:** Katherine S. Pollard.

**Data curation:** Svetlana Lyalina.

**Formal analysis:** Svetlana Lyalina.

**Funding acquisition:** Katherine S. Pollard.

**Investigation:** Svetlana Lyalina, Frank Wu, Shomyseh Sanjabi, Katherine S. Pollard.

**Methodology:** Svetlana Lyalina, Ramunas Stepanauskas.

**Resources:** Katherine S. Pollard.

**Supervision:** Ramunas Stepanauskas, Shomyseh Sanjabi, Katherine S. Pollard.

**Visualization:** Svetlana Lyalina, Katherine S. Pollard.

**Writing – original draft:** Svetlana Lyalina.

**Writing – review & editing:** Ramunas Stepanauskas, Shomyseh Sanjabi, Katherine S. Pollard.

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
