## [Decision Letter · Decision Letter 0]

28 Dec 2021

PONE-D-21-37230Single cell genome sequencing of laboratory mouse microbiota improves taxonomic and functional resolution of this model microbial communityPLOS ONE

Dear Dr. Pollard,

Thank you for submitting your manuscript to PLOS ONE. After careful consideration, we feel that it has merit but does not fully meet PLOS ONE’s publication criteria as it currently stands. Therefore, we invite you to submit a revised version of the manuscript that addresses the points raised during the review process.

We look forward to receiving your revised manuscript.

Kind regards,

Chih-Horng Kuo, Ph.D.

Academic Editor

PLOS ONE

Journal Requirements:

Reviewers' comments:

Reviewer's Responses to Questions

**Comments to the Author**

1. Is the manuscript technically sound, and do the data support the conclusions?

Reviewer #1: Partly

Reviewer #2: Yes

2. Has the statistical analysis been performed appropriately and rigorously? 

Reviewer #1: Yes

Reviewer #2: Yes

3. Have the authors made all data underlying the findings in their manuscript fully available?

Reviewer #1: Yes

Reviewer #2: No

4. Is the manuscript presented in an intelligible fashion and written in standard English?

Reviewer #1: Yes

Reviewer #2: Yes

5. Review Comments to the Author

Reviewer #1: This manuscript describes a single cell genomics approach for mouse microbiota in improving our current knowledge about taxonomic distribution and functional resolution. The topic should be very interesting; I however find that the manuscript does not explain everything very well, especially in the methodology part. Below I give my reviews.

1. The methodology of this manuscript is vague, especially in the computation part. For example, the manuscript mentioned that “selected the 150 SAGs from each sample that maximize the phylogenetic diversity and exclude SAGs with low probability of high genome recovery (Method).” The method however only refers to a python code without explaining anything, and the code is also vague at best (for example, what does the ‘costs file’ mean, and where one can obtain the Newick tree, etc.) As a result I have no idea in how the “maximization” is conducted and how the 150 SAGs from each sample are selected or determined. I of course can guess that the tree is probably from the GTDB-Tk results, but please make the methods as clear and reproducible by others as possible (just imagine someone who really want to try this approach based on this paper). Another example is how the assembly is conducted (Line 256 and beyond). I stress that these are NOT the only two places that have this problem so please look through the manuscript and add into methodology details as clear as you can.

2. How low is low-coverage and how high is high-coverage? I know that the mean reads number is 765,918, but what are the coverages after reads trimming? Please report coverages instead of reads count for both low and high coverages.

3. How one of the two samples of the wild type mice is selected? Random?

4. Method is too succinct. No description about how reads processing/trimming, assembly, and taxonomic lineage assignment were conducted. Please include the methodology even if the previous literature provides the description.

5. Line 292: DIAMOND should be all capital letters.

6. Please indicate software version numbers is available, for example checkM, GenomeTreeTk, or GTDB-Tk (I may not be comprehensive so please check it throughout the manusccript). If the authors also know the version of affiliating software version to the tools (e.g. pplacer for GTDB-Tk or DIAMOND for EggNOG-mapper), please also indicate them.

7. Line 90: “Further sequencing and assembly of DNA from the corresponding cells produced 298 high-coverage SAGs after quality control.” � again, please fill in details.

8. The study only talked about the increase of phylogenetic diversity of the Muribaculaceae and Bacteroidaceae families. But what about “f__Lachnospiraceae”? My impression from Figure 2 is that it is quite wide, but the authors did not even mention this family throughout the manuscript except in Figure 2. This also goes back to how the author “calculate” how much the current GTDB-Tk tree can potentially “gain” by adding the genomes. Please try clarifying these points.

9. This is up to the author’s discretion, but in my opinion I think Figure 3 are better described as Venn diagram instead of Euler diagram. I understand that for the plots in Figure 3 both diagram types are very similar. Maybe the authors can check out the definitions of both Venn and Euler diagrams and see if they agree with me.

10. Did the authors filtered the output results of CRISPRCasTyper, say whether the probability of the trustworthiness of the predicted CRISPRs to be real? I asked this question because there are CRISPR with only operons but no array. This question also reflects back to the previous question of describing the entire workflow as precise as possible, including the tools/parameters and what you do on handling or filtering.

11. There should be more to be discussed on CRISPR instead of just mentioning that Prevotella have both Type I and III systems in your data. For example, what benefit can the entire CRISPR analysis bring to this study? How the arrays are distributed within/between clades? These analysis should be able to bring merits to the manuscript and the associating genomes.

12. What does “hash” in Methods and figure S4/S5 means? If this means “k-mers” as I guessed, please just use “k-mer” or similar terms.

13. Also what are the y-axes in figure S4/S5 and how to interpret the figures? For example, for the most top-left figure “dnr” I can probably guess that the scg_ref lines indicates the reads or k-mer numbers that can be mapped to the database. But why distributions instead of just a number or two? And are the y-axes linear in a way that can connect different samples together? Please explain this figure as clear as you can.

14. There should be a way to depict the results represented in Figure S4/S5 into a clear message in the main manuscript. For example a few (but not too many) boxplots or violinplots that one can see very clearly that the combination of scg and current genome set can indeed classify better. Please consider adding a figure that says this message better.

Reviewer #2: The authors acquired the genomes of the novel bacterial species in gut microbiota of laboratory mice by single cell genome sequencing. It seems that the authors achieved expansion of the genome database about mice gut microbiota. However, some additional information may be required to support the author’s claim.

Though the SAGs were selected to maximize phylogenetic diversity, it seems that 70% of the SAGs derived from p__Bacteroidota and 40% of the SAGs derived from g__Prevotella from figure 3. Is it reflect the composition of microbiota? If not so, how the authors think what is the cause of taxonomical bias of the single-cell genomics? The taxonomical bias may be a limitation of improving taxonomic resolution by single-cell genomics.

(Fig. 1, S1) Please add the histogram or box plot of contamination to the figures.

(Line 70) Which is the evidence for the text that the range of GC% across the assemblies became wider by the WGA-X? If the ratio of bacteria containing genomes with ≧ 60 GC% is small, the range of GC% does not seem to change (from ref. 12).

(Line 119) Can the authors add the information about which SAG the new gene was obtained from? If the novel genes were acquired from SAGs of novel bacterial species, it may highlight that increasing the number of mouse gut species improve the microbial community analysis.

(Line 135) What is the meaning of “substantial novelty” in the title of Fig 3.

(Line 152) “outer ring of Fig 3” → Fig 2?

(Line 153) It was not clear how Prevotella SAGs form a subcluster. If it means that the subcluster distinguish from the already known Prevotella genomes, please add the data about that. The presence of the CRISPR-cas system or BGC depends on the completeness of the SAG, so the completeness should be added to Fig. 2 if possible.

(Line 262) Please write the threshold of SAG filtration. Why were 2 SAGs excluded after quality control from the 300 SAGs in high-coverage sequencing?

6. PLOS authors have the option to publish the peer review history of their article (what does this mean?). If published, this will include your full peer review and any attached files.

Reviewer #1: No

Reviewer #2: No

---

## [Author Response · Author response to Decision Letter 0]

11 Feb 2022

These responses to reviewer comments are also attached as a file with colored fonts to denote review text versus our replies. 

Reviewer #1: This manuscript describes a single cell genomics approach for mouse microbiota in improving our current knowledge about taxonomic distribution and functional resolution. The topic should be very interesting; I however find that the manuscript does not explain everything very well, especially in the methodology part. Below I give my reviews.

1.1: The methodology of this manuscript is vague, especially in the computation part. For example, the manuscript mentioned that “selected the 150 SAGs from each sample that maximize the phylogenetic diversity and exclude SAGs with low probability of high genome recovery (Method).” The method however only refers to a python code without explaining anything, and the code is also vague at best (for example, what does the ‘costs file’ mean, and where one can obtain the Newick tree, etc.) As a result I have no idea in how the “maximization” is conducted and how the 150 SAGs from each sample are selected or determined. I of course can guess that the tree is probably from the GTDB-Tk results, but please make the methods as clear and reproducible by others as possible (just imagine someone who really want to try this approach based on this paper). Another example is how the assembly is conducted (Line 256 and beyond). I stress that these are NOT the only two places that have this problem so please look through the manuscript and add into methodology details as clear as you can.

We appreciate this advice about reproducible and clear methods and have added methodological details in many places throughout the results and methods. 

Regarding the specific suggestion to improve the technical description of the optimization procedure, we have added text explaining each of the components and how they are used. In addition, we now provide citations that make the link between our specific use case and the broad class of prioritization problems being investigated in conservation biology (Lines 449-453). For readers interested in trying out the python implementation, we have added example input data (costs file and tree) in the github repository. For SAG assembly, we have added detailed descriptions of the methods involved in getting analysis-ready SAGs, as well as specific options and tool versions used for downstream feature annotation.

1.2: How low is low-coverage and how high is high-coverage? I know that the mean reads number is 765,918, but what are the coverages after reads trimming? Please report coverages instead of reads count for both low and high coverages.

We have computed and inserted into the text median coverages for both sequencing runs (Lines 74-75 and Line 106).

1.3: How one of the two samples of the wild type mice is selected? Random?

Text has been updated to clarify that choice of mouse was indeed random (Line 436).

1.4: Method is too succinct. No description about how reads processing/trimming, assembly, and taxonomic lineage assignment were conducted. Please include the methodology even if the previous literature provides the description.

This is an important critique that we have thoroughly addressed with a new Methods section that is double the length of the prior one. The Methods section now includes technical details for steps that were not previously described, as well specific additional details about software, protocols, and experimental reagents that were added to existing methods paragraphs. These changes have greatly expanded the Methods with information that addresses all the concerns raised by the reviewer.

1.5: Line 292: DIAMOND should be all capital letters.

Capitalization has been corrected.

1.6: Please indicate software version numbers is available, for example checkM, GenomeTreeTk, or GTDB-Tk (I may not be comprehensive so please check it throughout the manusccript). If the authors also know the version of affiliating software version to the tools (e.g. pplacer for GTDB-Tk or DIAMOND for EggNOG-mapper), please also indicate them.

Software versions have been added to the tools cited, and their affiliated software has been cited as well.

1.7: Line 90: “Further sequencing and assembly of DNA from the corresponding cells produced 298 high-coverage SAGs after quality control.” again, please fill in details.

Greater detail on the entire pipeline has been added, including assembly generation and QC. The reviewer may be specifically concerned about the two high-coverage SAGs that were discarded due to not being singletons; that information has been added too.

1.8: The study only talked about the increase of phylogenetic diversity of the Muribaculaceae and Bacteroidaceae families. But what about “f__Lachnospiraceae”? My impression from Figure 2 is that it is quite wide, but the authors did not even mention this family throughout the manuscript except in Figure 2. This also goes back to how the author “calculate” how much the current GTDB-Tk tree can potentially “gain” by adding the genomes. Please try clarifying these points.

Thanks for this suggestion. We have added discussion of the Lachnospiraceae family (Lines 253-269) and have explicitly defined the term “phylogenetic gain” (Lines 248-249, 465-466) to alleviate any confusion.

1.9: This is up to the author’s discretion, but in my opinion I think Figure 3 are better described as Venn diagram instead of Euler diagram. I understand that for the plots in Figure 3 both diagram types are very similar. Maybe the authors can check out the definitions of both Venn and Euler diagrams and see if they agree with me.

We agree with the reviewer that “Venn diagram” is the more correct term and have changed the text accordingly, in addition to explicitly citing the R package used for this visualization.

1.10: Did the authors filtered the output results of CRISPRCasTyper, say whether the probability of the trustworthiness of the predicted CRISPRs to be real? I asked this question because there are CRISPR with only operons but no array. This question also reflects back to the previous question of describing the entire workflow as precise as possible, including the tools/parameters and what you do on handling or filtering.

We have rerun the CRISPR-Cas subtype prediction with the most up to date version of CRISPRCasTyper (v1.6.2, version number added to text) and have updated the results accordingly. To maximize the number of candidate feature annotations, we performed this at a more permissive confidence threshold (0.5, now noted in text), and have depicted the confidence scores for each prediction in Figure 2 via the transparency aesthetic in ggplot2 (a modification to Figure 2 from its earlier version). We believe this approach shows the most possible data, while drawing the reader’s eye to the more trustworthy predictions. Based on the additional test we performed comparing the completeness of SAGs with both operon and array versus those with operons only, we hypothesize that these cases are related to insufficient assembly.

1.11: There should be more to be discussed on CRISPR instead of just mentioning that Prevotella have both Type I and III systems in your data. For example, what benefit can the entire CRISPR analysis bring to this study? How the arrays are distributed within/between clades? These analysis should be able to bring merits to the manuscript and the associating genomes.

We thank the reviewer for this suggestion and their interest in our CRISPR-Cas analysis. To address this comment, we have added text summarizing and interpreting these results, as well as some potential future directions for information gained from this annotation. These include exploring the diversity of bacterial anti-viral systems, leveraging this diversity to discover new genome editing tools, and possibly decoding host-phage relationships. 

1.12: What does “hash” in Methods and figure S4/S5 means? If this means “k-mers” as I guessed, please just use “k-mer” or similar terms.

Hash is not entirely equivalent to individual kmers, as MinHash-style approaches (sourmash, Mash) rely on compressed sketches of large sets of kmers. These sketches are chosen to be representative of the original data, but the process is not lossless. The text has been amended to say “kmer hash” for it to be more familiar to a broader audience. A brief discussion of the distinguishing features of sourmash that may make its results hard to directly compare to MIDAS has also been added (Lines 211-216).

1.13: Also what are the y-axes in figure S4/S5 and how to interpret the figures? For example, for the most top-left figure “dnr” I can probably guess that the scg_ref lines indicates the reads or k-mer numbers that can be mapped to the database. But why distributions instead of just a number or two? And are the y-axes linear in a way that can connect different samples together? Please explain this figure as clear as you can.

Supplementary figures 4 and 5 are meant primarily to provide a visual alternative to the tabular result of assessing the differences in performance on the “re-mapping” task. The intent is to show the reader the data as fully as possible, including potential irregularities like skewness and bimodality, which would not be easily visible with single number summaries. Ridgeline plots are a particular case of faceted density plots that are allowed to overlap on the y-axis, necessitating the removal of the conventional axis label (“density”) and tick marks (numeric values of kernel density estimates of the probability distribution of a particular variable). We have added detail to the supplemental figure captions to clarify the unorthodox details of this plot type.

1.14: There should be a way to depict the results represented in Figure S4/S5 into a clear message in the main manuscript. For example a few (but not too many) boxplots or violinplots that one can see very clearly that the combination of scg and current genome set can indeed classify better. Please consider adding a figure that says this message better.

We have taken a subset of results depicted in Figures S4/S5 and extracted them to a new main figure (Fig 4), which contains boxplots and swarmplots of the MIDAS and sourmash comparisons for just the murine sample test scenarios, which benefit the most from the new reference sequences.

 

Reviewer #2: The authors acquired the genomes of the novel bacterial species in gut microbiota of laboratory mice by single cell genome sequencing. It seems that the authors achieved expansion of the genome database about mice gut microbiota. However, some additional information may be required to support the author’s claim.

2.1: Though the SAGs were selected to maximize phylogenetic diversity, it seems that 70% of the SAGs derived from p__Bacteroidota and 40% of the SAGs derived from g__Prevotella from figure 3. Is it reflect the composition of microbiota? If not so, how the authors think what is the cause of taxonomical bias of the single-cell genomics? The taxonomical bias may be a limitation of improving taxonomic resolution by single-cell genomics.

While the SAGs selected for final high-coverage sequencing were chosen to maximize phylogenetic diversity, the reviewer is correct that there is an initial upper bound on how much diversity can be present in two individual ecological sites. This would be improved by sampling more mice from different facilities, and likely even further by sampling wild mice or even more interesting lab mice (like the “wildlings” from the work of Rosshart et al). As the work presented in this manuscript was intended as a proof of concept, we expect future contributions by the scientific community to surpass it in the amount of diversity captured. We have added text to the discussion further underscoring this informational bottleneck and outlining possible next directions for further exploration (Lines 330-337).

2.2: (Fig. 1, S1) Please add the histogram or box plot of contamination to the figures.

Figure S1 has been augmented to include contamination percentages as boxplots (panel B). As there was a relatively low threshold for allowable contamination, the range of numbers for this metric is quite small. 

2.3: (Line 70) Which is the evidence for the text that the range of GC% across the assemblies became wider by the WGA-X? If the ratio of bacteria containing genomes with ≧ 60 GC% is small, the range of GC% does not seem to change (from ref. 12).

This is a statement about the technology itself (covered by ref 12), not about the specific sample sequenced here (i.e., there was no replicate experiment performed with MDA, as it has been shown already that WGA-X is a better approach). We have added text to clarify that this statement is a reasonable supposition given the data, and an additional experiment would be required to fully prove this.

2.4: (Line 119) Can the authors add the information about which SAG the new gene was obtained from? If the novel genes were acquired from SAGs of novel bacterial species, it may highlight that increasing the number of mouse gut species improve the microbial community analysis.

To investigate this possibility, we re-extracted the SAG ids from the results of CD-HIT-EST-2D comparison against the Xiao et al (2015) mouse microbiome gene catalog. After tallying these data, we added the resulting number of novel ORFs from each SAG as an extra barplot annotation layer to the circular tree figure (Fig 2)

2.5: (Line 135) What is the meaning of “substantial novelty” in the title of Fig 3.

The wording “substantial” was used to not misleadingly claim that a formal statistical test was performed to determine the significance of the reported non-overlap percentages. Finding an appropriate test for this scenario is not straightforward and runs the risk of making overly strong assumptions about the baseline probability of a gene to fall into certain groups.

2.6: (Line 152) “outer ring of Fig 3” → Fig 2?

Typographic error due to figure rearrangement has been corrected.

2.7: (Line 153) It was not clear how Prevotella SAGs form a subcluster. If it means that the subcluster distinguish from the already known Prevotella genomes, please add the data about that. The presence of the CRISPR-cas system or BGC depends on the completeness of the SAG, so the completeness should be added to Fig. 2 if possible.

We have added completeness estimates as a barplot ring to Figure 2. We have also clarified that the “subcluster” being referred to is a qualitative determination in terms of features that distinguish this subset of SAGs versus the others. The astute caveat stated by the reviewer has been added to the text and an explicit test is performed on the CRISPR-Cas predictions to show that SAGs from the “operons only” group have lower completeness (Lines 179-185). This potential confounder is noted in both Results and Discussion. 

2.8: (Line 262) Please write the threshold of SAG filtration. Why were 2 SAGs excluded after quality control from the 300 SAGs in high-coverage sequencing?

Two SAGs were excluded due to high likelihood of them being doublets (i.e. sequences were not derived from single cells). Text has been edited to explicitly clarify this loss of 2 SAGs as well as how this final filtering was performed (Lines 107-110).

---

## [Decision Letter · Decision Letter 1]

2 Mar 2022

Single cell genome sequencing of laboratory mouse microbiota improves taxonomic and functional resolution of this model microbial community

PONE-D-21-37230R1

Dear Dr. Pollard,

We’re pleased to inform you that your manuscript has been judged scientifically suitable for publication and will be formally accepted for publication once it meets all outstanding technical requirements.

Kind regards,

Chih-Horng Kuo, Ph.D.

Academic Editor

PLOS ONE

Additional Editor Comments (optional):

Both reviewers and I are happy with the revised version, congratulations on this nice work.

Please note that Reviewer 1 identified a minor mistake in references. More info could be found here (https://cran.r-project.org/doc/FAQ/R-FAQ.html#Citing-R). I trust that this minor issue can be corrected by the authors and do not need another round of evaluation by me or the reviewers. Please work with the editorial office directly.

Reviewers' comments:

Reviewer's Responses to Questions

**Comments to the Author**

1. If the authors have adequately addressed your comments raised in a previous round of review and you feel that this manuscript is now acceptable for publication, you may indicate that here to bypass the “Comments to the Author” section, enter your conflict of interest statement in the “Confidential to Editor” section, and submit your "Accept" recommendation.

Reviewer #1: All comments have been addressed

Reviewer #2: All comments have been addressed

2. Is the manuscript technically sound, and do the data support the conclusions?

Reviewer #1: Yes

Reviewer #2: Yes

3. Has the statistical analysis been performed appropriately and rigorously? 

Reviewer #1: Yes

Reviewer #2: Yes

4. Have the authors made all data underlying the findings in their manuscript fully available?

Reviewer #1: Yes

Reviewer #2: Yes

5. Is the manuscript presented in an intelligible fashion and written in standard English?

Reviewer #1: Yes

Reviewer #2: Yes

6. Review Comments to the Author

Reviewer #1: The authors have successfully addressed my comments and concerns. I only want to point a citation that I did not catch in the first round of review. Citation [73] is incorrect in both the author name and the DOI. The correct citation should be as follows.

R Core Team (2016) R: A Language and Environment for Statistical Computing. R Foundation for Statistical Computing, Vienna, Austria.

https://www.R-project.org/

Reviewer #2: The authors have satisfactorily addressed previous comments and suggestions raised by the reviewers. I have no additional comments on the current version of the manuscript.

7. PLOS authors have the option to publish the peer review history of their article (what does this mean?). If published, this will include your full peer review and any attached files.

Reviewer #1: No

Reviewer #2: No

---

## [Editor Report · Acceptance letter]

31 Mar 2022

PONE-D-21-37230R1 

Single cell genome sequencing of laboratory mouse microbiota improves taxonomic and functional resolution of this model microbial community 

Dear Dr. Pollard:

I'm pleased to inform you that your manuscript has been deemed suitable for publication in PLOS ONE. Congratulations! Your manuscript is now with our production department. 

Kind regards, 

on behalf of

Dr. Chih-Horng Kuo 

Academic Editor

PLOS ONE